# Effects of Pulse Current Charging on the Aging Performance of Commercial Cylindrical Lithium Ion Batteries

Seunghun Lee [1], Wonil Cho [1], Vandung Do [1] and Woongchul Choi [2],*

1   Center for Energy Storage Research, Korea Institute of Science and Technology, 5, Hwarang-Ro, Seongbuk-Gu, Seoul 02792, Korea; t16806@kist.re.kr (S.L.); wonic@kist.re.kr (W.C.); dodv@kist.re.kr (V.D.)
2   Department of Automotive Engineering, Kookmin University, 77 Jeongneung-Ro, Seongbuk-Gu, Seoul 02707, Korea
*   Correspondence: danchoi@kookmin.ac.kr; Tel.: +82-02-910-5461

**Abstract:** Rapid development of electronic devices, ranging from personal communication devices to electric mobility solutions, has increased demand for energy storage devices not only in the production volume but also in the product functionality. Among many functional requirements including energy capacity, safety, and short recharge time, one of the major limitations is the short charging time while maintaining the designed capacity. However, even with the most updated lithium-ion battery (LIB) technology, it is well known that fast charging with a high current rate would reduce the lifetime of batteries significantly. Recently, among the many approaches to improve the quick charging performance, a pulse current charging method while keeping the total amount of energy has demonstrated a successful fast recharging of LIB without significantly degrading the battery capacity. The essence of the idea is to stop charging in the middle stage to provide a relaxation period instead of continuously charging at a high current rate. In this study, a comparative study between a conventional charging method with 3C current rate (equivalent to 20 min of charging time) and a pulse current charging with 6C current rate (10 min of charging and 10 min of relaxation time) was carried out. While the conventional charging method showed that the capacity was maintained up to about 200 cycles, the pulse current charging method revealed that the capacity was maintained for more than 450 cycles with a Coulombic efficiency of nearly 100%.

**Keywords:** li-ion battery; fast charging; state of health (SOH); solid electrolyte interface (SEI); aging; pulse current charging

## 1. Introduction

The appearance of the commercial lithium-ion battery (LIB) in 1991 [1] has opened a new era for the boom in electronic and mobile devices, leading to the participation in social networks as well as the electrification of automobiles. The LIB has become an essential part of smartphones as well as electric vehicles (EVs), helping to increase human connection, expand the capabilities of electronic devices, and, eventually, enable the reduction of the use of fossil fuels. With the dire concerns about climate changes due to greenhouse gas emissions from the combustion engine vehicles [2], EV has become a promising solution to replace the conventional vehicles with the support of the LIB [3]. The continuous increase in the role of the LIB has led to further improvements and enhancements of the battery performance. Especially for the successful deployment of EV, it is critical to minimize the charging time of the LIB so that users feel comfortable with the refueling of vehicles at charging stations [4]. To improve user convenience through the fast charging of EV, two different aspects have to be improved together. Firstly, the infrastructures of EV service equipment, namely the chargers and related electric supply facilities, have to be developed to be able to supply high enough current rate to the EV battery pack. This is known to be relatively simple compared to the improvement of battery functionality itself. Secondly, the chemistry of the LIB, namely the battery functionality itself, has to be further refined to be

able to accommodate the higher current charging while keeping the battery capacity without significant degradation over the acceptable lifecycle of the EV applications. Unlike the case of EV service equipment development, this battery chemistry refinement for the rapid charging speed is known to be extremely challenging [5]. To fully understand the charging process of the LIB and possibly improve the performance in that regard, three major processes are to be investigated carefully: (i) rapid lithium de-intercalation from the positive electrode; (ii) rapid diffusion of lithium through the electrolyte; and (iii) rapid lithium insertion into the negative electrode. In fact, the complexity of designing and manufacturing the LIB makes it very difficult to satisfy all of these factors simultaneously [6].

In the current research, our team paid close attention to the third aspect, the rapid lithium insertion into the negative electrode because the solid electrolyte interface (SEI) definitely experiences build up during the repeated charging processes and is well known to be a major mechanism of battery capacity degradation. More importantly, the frequency of charging pulse was carefully considered for its possible contribution to maintaining the battery health through minimizing the growth of the SEI. To understand and explain the fast-charging acceptance of a mass-produced LIB by applying the pulse current charging, the change in battery performance was investigated by observing its lifecycle, as well as using electrochemical impedance spectroscopy (EIS) [7].

## 2. Experimental Method

An INR18650-29E cylindrical battery manufactured by Samsung SDI, which has a nominal capacity of 2850 mAh, was used to investigate the battery performance by the pulse current charging. The materials used in INR18650-29E cylindrical batteries are NCM111 for cathode and graphite for anode. To precisely control the charging and discharging processes over many cycles, a Maccor battery cycler (series 4000) was used. All charging/discharging processes were conducted in room temperature test condition (26 °C).

For electrochemical impedance analysis, an electrochemical impedance spectrum (EIS) analyzer, SI 1260 Impedance/Gain-Phase Analyzer from Solartron, and SI 1286 electrochemical interface, also from Solartron, were connected to the Maccor battery cycler, as illustrated in Figure 1.

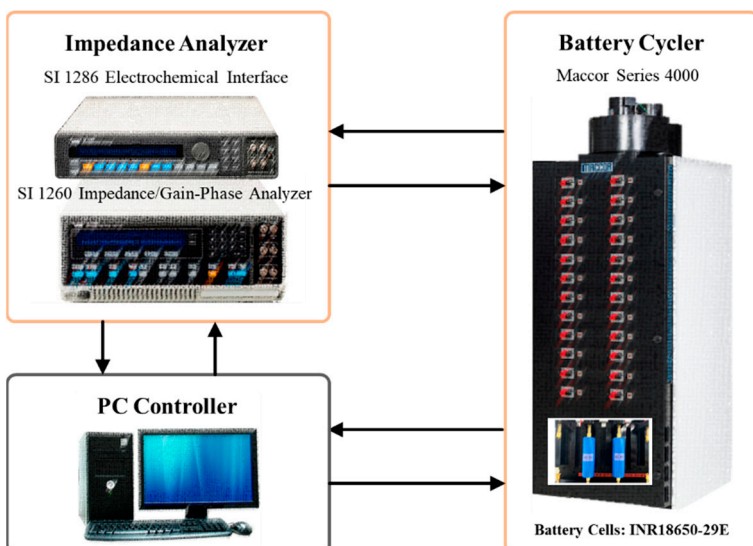

**Figure 1.** Experimental setup.

The reference experiments were carried out with the constant current charging at 3C, while the comparative experiments were conducted with the selected pulse current charging at 6C, i.e., repeated applications of 6C charging and relaxation, resulting in the same total amount of energy supplied as the reference experiment. For the discharging processes, the 1C current rate was kept the same for all cases. For all cases, a rest time of

5 min was given between each process, i.e., 5 min of rest time after charging and 5 min of rest time after discharging.

In all experiments, batteries were conditioned before the initiation of the actual charging and discharging experiments. First, batteries were discharged completely. After the complete discharge, the batteries were charged once at 0.1C (0.28 A) to the 20% state of charge (SOC). With the successful initialization of the batteries, the charging and discharging processes were the, repeated between 20% SOC and 50% SOC. In this study, experiments were carried out with the SOC defined as $Q_C/Q_I$, where $Q_C$ is the current capacity and $Q_I$ is the initial capacity of the battery prior to any degradation. As shown in Figure 2, for the case of a 3C charging and 1C discharging set, the charging process used a constant current of 8.4 A (3C) with the 4.2 V charging voltage limit. The discharging process used a constant current of 2.8 A (1C) applied under the condition of the discharge limit voltage of 3.0 V. For the case of a 6C pulse charging and 1C discharging set, the charging process used a pulse current of 16.8 A (6C) with the charging voltage limit of 4.2 V. In other words, 16.8 A (6C) were applied for 0.5 s followed by the relaxation period of 0.5 s, repeatedly, which resulted in the same amount of energy charged to the battery. The frequency of the charging process was selected based on the observation of EIS test results from the batteries used in the current study. The discharging process used a constant current of 2.8 A (1C) applied under the condition of the discharge limit voltage of 3.0 V, as in the case of 3C charging, the reference case. For both cases, a resting period of 1 min was set after the completion of each cycle. After the 1 min of resting period, the next cycle with the specified current profile was repeated. Electrochemical impedance spectra of the batteries were measured at 0% SOC every 50 cycles using SI 1260 Impedance/Gain-Phase Analyzer (Solartron) and SI 1286 electrochemical interface (Solartron) connected to the Maccor battery cycler, as shown in Figure 1.

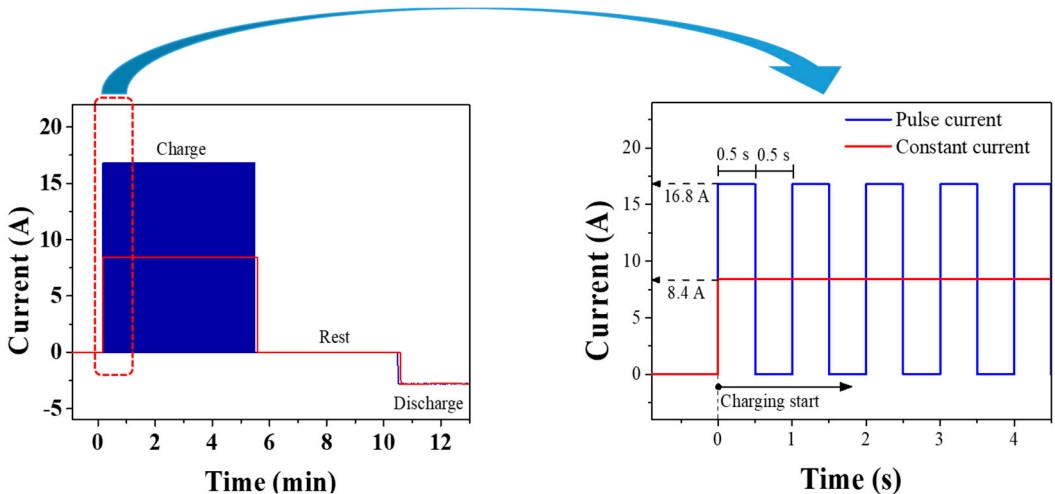

**Figure 2.** Charging current-time profile of constant current and pulse current procedures.

## 3. Results and Discussion

To understand the effect of the rapid charging at 3C with a conventional constant current profile (3C-CC) and the rapid charging at 6C with a pulse current profile (6C-PC) on the performance of the commercial cylindrical LIB, the charging and discharging conditions were kept the same as explained above. Figure 3 shows the discharge capacity and Coulombic efficiency according to the number of charge and discharge cycles. In the case of the 3C constant current charging, the capacity retention and the Coulombic efficiency were maintained stable during the first 50 cycles, and then gradually decreased over the next 150 cycles. Considering the fact that the rapid charging was carried out over the range from 20% SOC to 50% SOC, which is believed to be a rather robust, safe, and stable range against charging stress and therefore does not significantly affect the

battery performance, there was a clear fluctuation in both the capacity and Coulombic efficiency after 50 cycles. It was believed that the high charging current value of 8.4 A for relatively longer time might cause battery deterioration. In contrast to the constant current charging, the pulse charging, which was constructed with 0.5 s of 6C current charging and 0.5 s of resting time, astonishingly showed stable capacity and Coulombic efficiency over 450 cycles.

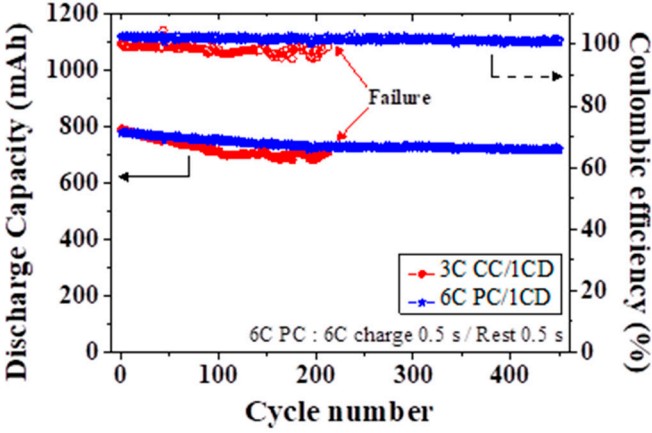

**Figure 3.** Discharge capacity and Coulombic efficiency.

To clarify the difference between the two charging methods, the capacity–voltage profiles were investigated, as shown in Figure 4a for the case of 3C constant current charging and Figure 4b for the case of 6C pulse current charging. In the case of 3C constant current charging, the initial slope of the charging voltage curve became steep as the cycle progressed, and the increase in voltage magnitude at the end of charging was noticeably observed. In particular, on the 150th charging voltage curve, it can be seen that a very small capacity was obtained during constant current charging by the charge voltage limit of 4.2 V. This might come from the over-potential formation and lithium plating that occurred on the surface of the graphite anode, while the insertion process took place incompletely in the graphite structure due to the high C-rate charging [8,9].

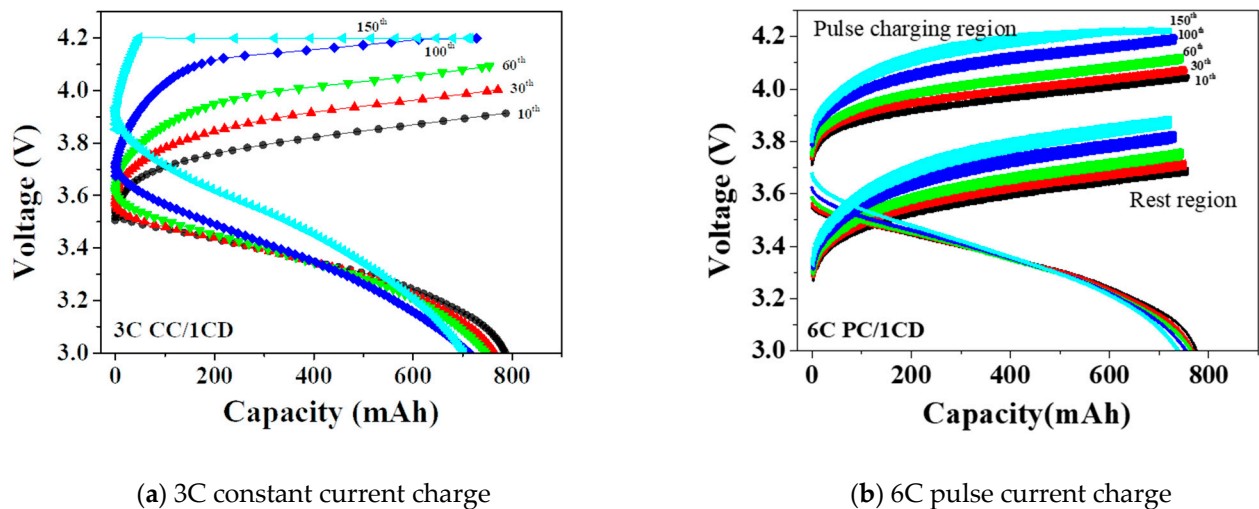

(**a**) 3C constant current charge ⠀⠀⠀⠀⠀⠀⠀ (**b**) 6C pulse current charge

**Figure 4.** Voltage-capacity profile at selected cycles.

It is known that the plated lithium interacts with the electrolyte to form the SEI, which increases the internal resistance of the battery. In addition, continuous SEI formation would result in lithium consumption, formation of "dead lithium" and lithium dendrites,

electrode cracking, and loss of electrical contact in electrodes [10,11], which increase the internal resistance of the battery and further decrease its capacity. Particularly in the current research, even though the repetitive charging and discharging were performed only within a portion (20–50% SOC) of the total battery capacity, the capacity decrease and the increase in charging start voltage (10–100th cycle, 0.185 V) clearly revealed the limitation of high C-rate constant charging on typical lithium ion batteries.

On the other hand, the 6C pulse current charging showed a voltage–capacity curve that was quite different from the 3C constant current charging one, as illustrated in Figure 4b. Charging was performed by switching between the high voltage and low voltage parts according to the pulse current application and the pause. In the current study, the pulse current was applied for 0.5 s and the pause was made for 0.5 s in sequence, as shown in Figure 2. Both high and low voltage curves were obtained from the battery under the current application and the pauses. As shown in Figure 4b, the voltage increment due to the progress of the charge and discharge cycle was relatively smaller than the one for 3C constant charging procedure, and a major amount of charging capacity was obtained before reaching the charging voltage limit of 4.2 V. This might be explained by the pulse current charging effectively mitigating the concentration polarization caused by the application of high current during the resting time of pulse charging [12]. The relaxation of concentration polarization during the resting period of pulse charging might have provided enough time for the diffusion and insertion reaction of lithium ions to proceed, and thus naturally suppressed the electrode over-potential and lithium plating, thereby reducing side reactions and consumption of active materials.

One more thing to clarify in Figure 4 is the discharging voltage levels of the battery at different cycles. For the case of an old battery, namely the battery after 150 cycles, due to the increase of internal resistance, the charging voltage increased relatively higher (near 4.2 V). After the 5 min of rest time, the discharging was initiated. With this limited rest time, the voltage did not reach down to an open circuit voltage. In other words, the discharging was initiated at the high voltage level of around 3.7 V before it really reached down to the fully converged open circuit voltage. By comparison, for the case of a fresh battery, i.e., the battery after 10 cycles, due to the low internal resistance, the charging voltage increased only up to 3.9 V. With the limited rest time of 5 min, the voltage dropped to near 3.62 V and then the discharging was initiated. Normally, a battery with higher internal resistance shows lower discharging voltage level. However, in the current study, the discharging processes were started only after a limited rest period, and therefore the voltage of the old battery was higher than the relatively fresh battery since the discharging started from the higher voltage level.

Figure 5 provides the charging start voltage (left $y$-axis) and the charging capacity (right $y$-axis) obtained from the 3C constant current charging method and the 6C pulse charging method. In terms of the voltage, the charging start voltage of the 6C-PC charging method (blue star marked line) in the initial 150 cycles shows a higher value than that of the 3C-CC charging method due to applying the higher input current charging procedure. In the case of 3C-CC charging method (red disk marked line), the charging start voltage increases about 0.185 V for the first 150 cycles, which is 62% higher than the corresponding value in the case of the 6C-PC charging method (0.071 V). With this observation, it is believed that the increase of internal resistance during the 3C constant current charging was greater than that of the 6C pulse current charging. Since the impedance increase in the LIB was mostly due to battery degradation, it is naturally believed that the 3C constant current charging process caused more battery performance deterioration than the 6C pulse current charging one. In Figure 5, the total charging capacity ($Q_{total}$) is indicated by a hatched bar, and the capacity charged before reaching the upper limit voltage ($Q_v$) is indicated by a filled bar. The charging capacity by the 3C constant current charging appeared somewhat higher up to 60 cycles than that by the 6C pulse current charging. However, as the cycles repeated, for the case of 3C constant current charging, the charging capacity at the 150th cycle was down to 715.62 mAh, which showed about 90% compared

to the charging capacity at the 10th cycle, and the corresponding $Q_v$ was found to be only 44.52 mAh. On the contrary, for the case of 6C pulse current charging, 723.58 mAh was charged by the pulse charging in the same cycle, recording about 95% of the charging capacity of the 10th cycle, and 546.14 mAh was charged before reaching the limit voltage of 4.2 V. As explained above, the pulse current charging on the lithium-ion battery caused relatively little increase in resistance and decrease in capacity of it, and the results support the aforementioned deterioration mitigation. It was confirmed that the charging proceeds very stably by repeated voltage drop due to the pulse current even after the 4.2 V limit voltage, which is explained again by comparing the charging time in Figure 6.

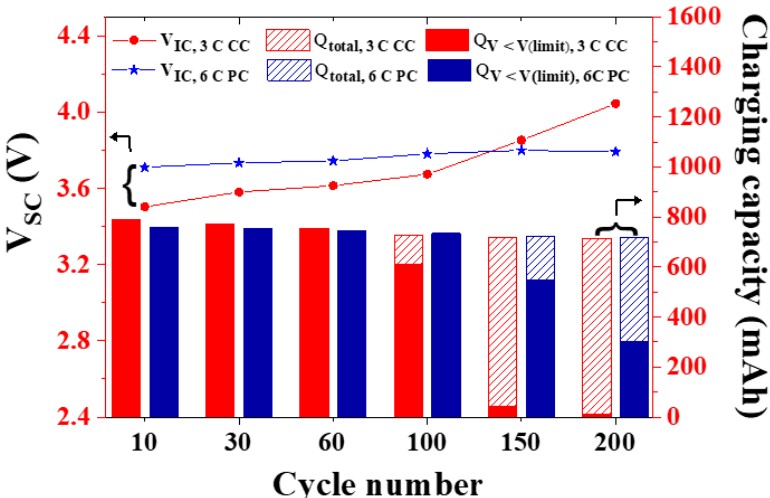

**Figure 5.** Charging start voltage and capacity in accordance with charging methods.

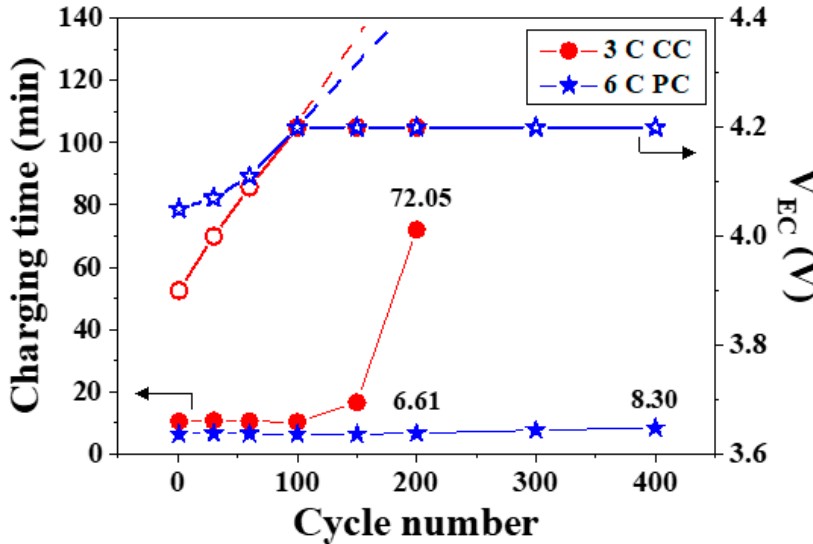

**Figure 6.** End voltage and elapsed time for each method.

As mentioned above, in the current research, the charge and discharge test of the commercial cylindrical battery was designed so that the amount of charging was limited to 3 Wh. This amount of current applied per hour is considered to allow the same capacity and to maintain the SOC range theoretically. The comparison of charging time in Figure 6 helps infer the effectiveness of the charging method on the battery. In the case of the 3C constant current charging on the battery, it appeared that the electrochemical insertion and desorption reactions of lithium are limited due to the deterioration caused by various factors, as discussed above. In addition, it showed that the charging time increased quite

rapidly as the cycle increased [13]. Specifically, it took about 10 min 30 s for the initial cycling of the battery, but it drastically increased after the 100th cycle and took about 72 min for the 200th charging process. After that, the battery reached a state where it was no longer possible to charge it in the 213th cycle. On the contrary, in the case of 6C pulse current charging, the initial charging took 6 min 20 s, while the 400th charging took approximately 8 min 20 s. These values for the charging process revealed that they did not deviate significantly from the theoretical time required to charge 30% of the full capacity in an ideal environment, which was estimated to be around 6 min. Since there was only about a 0.3 s/cycle increase in the charging time up to the 400th cycle, it could be interpreted as the pulse current charging was very effective, not only for fast charging but also for maintaining the power rate while keeping the stable battery operation. The end voltage of the charging increased from 3.91 to 4.21 V at 3C and converged to 4.2 V as the number of cycles increased, because it was the fixed value as charge limit voltage. The end voltage of 3C constant current charging reached the charge limit voltage before 100 cycles, displaying a very steep slope. On the other hand, in the 6C pulse current charging method, the end voltage gradually increased. The extrapolated lines of the ending voltage after reaching 4.2 V implied the slope of 3C constant current charging was steeper than that of 6C pulse current charging because the internal resistance due to battery degradation was larger under the 3C constant current charging case.

Finally, EIS measurements were performed after every 50 cycles for both cases to compare the changes of internal impedance, as presented in Figure 7a,b. After the end of first cycle (the first charging and discharging), the impedance spectra were similar. However, in the case of 3C constant current charging, the impedance curves, which were measured at the end of 1, 50, 100, and 150 cycles, showed great increase in their values depending on the number of cycles. The equivalent circuit model [14] constructed based on the result from the EIS measurement can be expressed as the electrolyte resistance ($R_s$), the first semicircle ($R_1$ and $C_1$), the second semicircle ($R_2$ and $C_2$), and the Warburg impedance ($W_0$). The main cause of the increase in charge transport resistance ($R_{ct}$) was that the lithium, which was plated with incessant concentration polarization, continuously reacted with the electrolyte under overvoltage to form SEI and consumed the electrolyte and lithium ions at the same time [15]. Furthermore, as the cycles repeated, the electrolyte resistance ($R_S$) gradually increased, and the overall curve shape was shifting to the right, which is believed to be a combination of electrolyte decomposition, aging, and electrode and separator deterioration due to repeated charging and discharging [15,16]. In contrast, in the case of the battery to which 6C pulse current charging was applied, the results show nominally small increases of the electrolyte resistance ($R_s$) and the charge transport resistance all the way up to 400 cycles, and these results clearly prove that the high C-rate pulse current charging method facilitated nice and smooth lithium insertion and diffusion into the graphite anode [17] under the testing condition. In other words, even though the high current level of 16.8 A was applied to the battery for 0.5 s, it was less damaged because there was a resting period of 0.5 s for relaxed lithium insertion and diffusion into the anode. Similar observations were reported by other researchers as well, while the viewpoints were somewhat different among each other. In the study by Li, J. et al. [12], SEM images of cathodes and anodes from all tested cells were taken, and the comparison showed that the cathodes were all in similar conditions regardless of charging profiles, whether conventional CC-CV or pulse charging, while the anodes showed significantly less SEI formation for the case of the pulse charging profiles. These results are nicely in line with the current study of 1 Hz pulse charging strategy, which specifically targeted minimal SEI formation during the charging. In addition, Abdel-Monem, M. et al. [18] compared multi-stage constant current and constant voltage (MCC-CV), conventional CC-CV, and pulse charging profiles. They concluded that the battery deterioration was severe for the case of CC-CV, while the rate of capacity decrease was minor for the cases of MCC-CV and the pulse charging and somewhat similar between them. However, they also reported that the lower rate of impedance rise was a clear characteristic of the pulse charging case

compared to the case of MCC-CV, which indicated the better anti-aging performance of the pulse charging strategy. Similar results has also been reported in other studies [18–21].

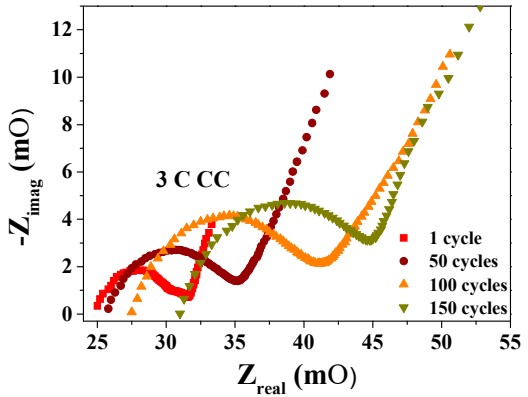

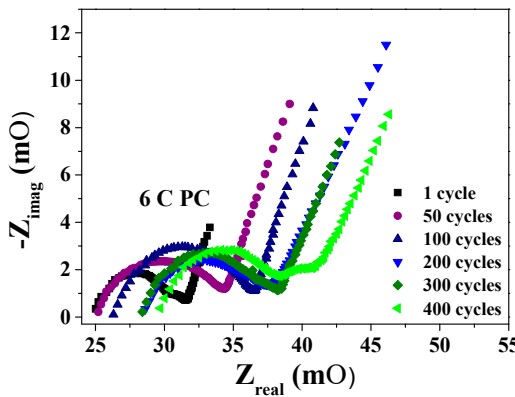

(**a**) 3C constant current charge and 1C discharge

(**b**) 6C pulse charge and 1C discharge

**Figure 7.** Electrochemical impedance spectrum (EIS) depending on charging methods.

## 4. Concluding Remarks

In the current research, a systematic approach to develop a high current fast charging algorithm for LIB and understand the mechanism underlying aging phenomena scenarios was carried out. Based on the comparative results, the 6C-PC charging method that contains 1 Hz algorithm (consisting of 0.5 s of charging and 0.5 s of resting) shows incredible electrochemical performance over 400 cycles with the targeted commercial cylindrical battery. The pulse charging method can support 30% charging of the battery capacity in 6 min 20 s, with a time offset of 0.3 s/cycle, which is much better than the traditional constant current charging method. In addition, the constant charge limit voltage (at 4.2 V) of the 6C-PC charging method implies that the method can greatly reduce the development of internal impedance. The EIS observation by cycling proves that the pulse current charging process limits the growth of the RS and the RCT, which can maintain well the lithium-ion insertion and diffusion into the electrodes. Astonishingly, this approach provided notable success in the fast charging with the pulse current charging. Therefore, the high-current pulse charging process of about 1 s is, in our opinion, very effective for commercial batteries and practical applications on EV.

**Author Contributions:** Conceptualization, W.C. (Wonil Cho) and W.C. (Woongchul Choi); methodology, S.L. and V.D.; investigation, S.L. and W.C. (Wonil Cho); formal analysis, W.C. (Wonil Cho) and W.C. (Woongchul Choi); resources, V.D.; validation, S.L. and W.C. (Wonil Cho); visualization, S.L. and V.D.; writing—original draft preparation, V.D. and W.C. (Woongchul Choi); and writing—review and editing, W.C. (Woongchul Choi) All authors have read and agreed to the published version of the manuscript.

**Funding:** This work was supported by the Korea Institute of Science and Technology Institutional Program (Project no., 2E30982) and also supported by Kookmin University Research Program (2020-0005).

**Institutional Review Board Statement:** Not applicable.

**Informed Consent Statement:** Not applicable.

**Data Availability Statement:** Not applicable.

**Conflicts of Interest:** On behalf of all authors, the corresponding author states that there is no conflict of interest.

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
