# Peer review of "Effects of Pulse Current Charging on the Aging Performance of Commercial Cylindrical Lithium Ion Batteries"

_applsci, doi:10.3390/app11114918_

Round 1
Reviewer 1 Report
- How did your results address the problem?
- It is better to establish specific objectives Provide more details about the methodology
- It would be good to include in the discussion other similar experiments to confirm experimentation before the conclusions
- It is better to provide a more up-to-date bibliographic review. There are excellent, very current examples published on the subject of Information Journal. The article has no reference from the journal Information.
- Journal Information has excellent references related to the topic.
Author Response
Response to the comments from Reviewer 1
First of all, I really appreciate careful review from Reviewer 1. Based on the comments from Reviewer 1, I have revised the manuscript accordingly as follows.
Comment 1
How did your results address the problem?
Response 1
I truly appreciated the comment from Reviewer 1. I have revised (in blue color) the Concluding Remarks as follows.
In the current research, the systematic approach to develop a high current fast charging algorithm for LIB and understanding the mechanism underneath the aging phenomena scenarios were carried out. Based on the comparative results, the 6C-PC charging method that contains 1 Hz algorithm (consist of 0.5 sec charging and 0.5 sec resting) show incredible electrochemical performance over 400 cycles with the targeted commercial cylindrical battery. The pulse charging method can support for charging 30% of the battery capacity in 6 min 20 sec, with the time offset of 0.3 sec/cycle, which are much better than the traditional constant current charging method. In addition, the constant charge limit voltage (at 4.2 V) of the 6C-PC charging method implies that the method can greatly reduce the development of internal impedance. The EIS observation by cycling proves that the pulse current charging process limits the growth of the RS and the RCT, which can maintain well the lithium ion insertion and diffusion into the electrodes. Astonishingly enough, this approach provided the notable success in the fast charging with the pulse current charging. Therefore, the high-current pulse charging process of about 1 sec is, in our opinion, very effective for commercial batteries and practical applications on EV.
Comment 2
It is better to establish specific objectives. Provide more details about the methodology
Response 2
I really appreciated the comment from Reviewer 1. I have added a paragraph (in blue color) to provide a more specific objectives as follows in 1. Introduction and 2. Experimental Method.
- Introduction
In the current research, our team paid closed attention to the third aspect, (iii) the rapid lithium insertion into the negative electrode, because the solid electrolyte interface (SEI) which was known to be built up through repeated charging processes and known to be the major mechanism of battery capacity degradation. More importantly, a frequency of charging pulse was carefully considered for its possible contribution to maintaining the battery health through minimizing the growth of the SEI. In order to understand and explain the fast charging acceptance of a mass-produced LIB by applying the pulse current charging, the change in battery performance is investigated by observing the cycle life, as well as studying the electrochemical impedance spectroscopy (EIS) [7].
- Experimental Method
As shown in Figure 2, for the case of 3C charging and 1C discharging set, the charging process used a constant current of 8.4A (3C) with the 4.2 V charging voltage limit. The discharging process used a constant current of 2.8A (1C) applied under the condition of the discharge limit voltage of 3.0V. For the case of 6C pulse charging and 1C discharging set, the charging process used a pulse current of 16.8A (6C) with the charging voltage limit of 4.2 V. In other words, 16.8A (6C) was applied for 0.5 seconds followed by the relaxation period of 0.5 seconds, repeatedly which resulted in the same amount of energy charged to the battery. The frequency of the charging process was selected based on the observation of the EIS test results from the batteries used in the current study. The discharging process used a constant current of 2.8A (1C) applied under the condition of the discharge limit voltage of 3.0V just like the case of 3C charging, the reference case.
Comment 3
It would be good to include in the discussion other similar experiments to confirm experimentation before the conclusions.
Response 3
Again, I truly appreciated the comment from Reviewer 1. I have added a few paragraphs (in blue color) to discuss more with the various findings from other researchers as follows at the end of 3. Results and Discussion
Similar observations were reported by other researchers as well while the viewpoints were somewhat different among each other. In the study by Li, J. et al. [18], SEM images of cathodes and anodes from the all tested cells were taken and the comparison showed that the cathodes were all in the similar conditions regardless of charging profiles whether it is a conventional CC-CV or a pulse charging profile, while the anodes showed significantly less SEI formation for the case of the pulse charging profiles. These results were nicely in line with the current study of 1 Hz pulse charging strategy which targeted specifically for minimal SEI formation during the charging. Also, in the study by Abdel-Monem, M. et al. [19], comparison study between a multi-stage constant current and constant voltage (MCC-CV), a conventional CC-CV, and a pulse charging profiles were reported. The conclusion showed that the battery deteoration were severe for the case of CC-CV while rate of capacity decrease were minor for the cases of MCC-CV and the pulse charging and somewhat similar between them. Hoewever, they also reported that the lower rate of impedance rise were clear characteristic of the pulse charging case compared to the case of MCC-CV which implied that better anti-aging performance of the pulse charging strategy. Similar results were also reported in other studies as well [20-22].
Comment 4
It is better to provide a more up-to-date bibliographic review. There are excellent, very current examples published on the subject of Information Journal. The article has no reference from the journal Information.
Response 4
I fully agree with this comment from Reviewer 1. I have added more references accordingly as shown in the revised manuscript.
- Li, J., Murphy, E., Winnick, J. & Kohl, P. A. The effects of pulse charging on cycling characteristics of commercial lithium-ion batteries. J. Power Sources 2001, 102.
- Abdel-Monem, M., Trad, K., Omar, N., Hegazy, O., Van den Bossche, P. & Van Mierlo, J. Influence analysis of static and dynamic fast-charging current profiles on ageing performance of commercial lithium-ion batteries. J. Energy 2017, 120
- Chen, L. R., Wu, S. L., Shieh, D. T. & Chen, T. R. Sinusoidal-ripple-current charging strategy and optimal charging frequency study for Li-ion batteries. IEEE Trans Ind Electron 2013. 60(1)
- Amanor-Boadu, J. M., Guiseppi-Elie, A. & Sanchez-Sinencio, E. Search for optimal pulse charging parameters for Li-ion polymer batteries using Taguchi orthogonal arrays. IEEE Trans Ind Electron 2018. 65(11)
- Yin, M., Cho, J., Park, D., Yin, M. D., Cho, J. & Park, D. Pulse-based fast battery IoT charger using dynamic frequency and duty control techniques based on multi-sensing of polarization curve. Energies 2016. 9(3)
I DO appreciate the comments and suggestions from Reviewer 1 and I would like to express my sincere gratitude.

Reviewer 2 Report
This paper proposed a pulsed charging method for fast charging and save the battery high capacity and the author validated his points based on one experiment test. However, the experimental method is not rigorous, and the results have conflicted parts to the author's points. In addition, the proposed method lacks scientific analysis to explain the results.
More detailed comments as following:
Line 53: What are the materials used in the INR18650-29E cylindrical battery?
Line 68: why use the SOC range instead of the voltage range for the cycling? Would it be better to fully charge the battery and then fully discharge the battery each cycle to clearly understand the behavior? In addition, how this SOC is measured? The SOC can be calculated as Qc/Q, Qc is current capacity and Q is the battery total capacity. However, the battery's total capacity is changing during cycling due to the capacity fade.
Figure 4:
(1)It is hard to understand the figure. For both 4a and 4b, the voltage starts from 3.5V to 3.8V for charging, and then discharged from 3.5V to 3V for all cycles? Should be discharged from 4.2V to 3V and change from 3V to 4.2V?
(2)For 4a, it seems like the battery are not charged to the same condition, i.e. the stop charge voltage are significant differences for each cycle.
(3)For 4b, based on the author’s description, the battery is charged for 0.5s and relax for 0.5s. If this is true, why each charge/relax curve is so smooth and all start from almost the same voltage?
(4) In addition, the battery resistance increased after cycling, why the ohmic drop at the 150th cycle discharge curve is less than the ohmic drop at the 10th cycle discharge?
Line 147: Why it limited to 3Wh? This paper compared the 3C-CC and 6C-PC 0.5s pulse with 0.5s relaxation. Is 3C-PC or 10C-PC will be better than 6C-PC? Will 0.2s pulse better?
Author Response
Response to the comments from Reviewer 2
First of all, I really appreciate careful review from Reviewer 2. Based on the comments from Reviewer 2, I have revised the manuscript accordingly as follows.
Comment 1
Line 53: What are the materials used in the INR18650-29E cylindrical battery?
Response 1
I truly appreciated the comment from Reviewer 2. I have added a paragraph (in green color) to accommodate the comment as follows in 2. Experimental Method.
The materials used in INR18650-29E cylindrical battery are NCM111 for cathode and graphite for anode.
This cylinder cell is developed and manufactured by Samsung SDI. We cannot arbitrarily disassemble the cell. Likewise, it cannot be analyzed or published for the specific information of internal electrode materials ratio or electrolytes composition. Because a commercial cylinder cell was used, we were able to conduct experiments using a higher current than a lab-scale coin cell or pouch cell, and this was meaningful to evaluate the comparative study between a constant current and pulse current charging profiles from a practical point of view.
Comment 2
I appreciated careful comments from Reviewer 2. I have added a paragraph (in green color) to accommodate the comment as follows in 2. Experimental Method.
Line 68: why use the SOC range instead of the voltage range for the cycling? Would it be better to fully charge the battery and then fully discharge the battery each cycle to clearly understand the behavior? In addition, how this SOC is measured? The SOC can be calculated as Qc/Q, Qc is current capacity and Q is the battery total capacity. However, the battery's total capacity is changing during cycling due to the capacity fade.
Response 2
First, I noticed that the SOC definition was not clear explained in the manuscript. In this study, we experimented with the SOC defined as QC/QI, where QC is the current capacity, and QI is the initial capacity of the battery prior to any degradation. Accordingly, I revised the manuscript to include a statement as follows.
In this study, experiments were carried out with the SOC defined as QC/QI, where QC is the current capacity, and QI is the initial capacity of the battery prior to any degradation.
Further answers:
Because the charging voltage limit is reached very quickly due to the high current density (charging batteries at a high C-rate), a comparison of high speed charging methods becomes difficult without specifying an appropriate SOC range. In general, the constant current-constant voltage (CC-CV) current profile is applied to charge a battery to its full capacity. However, with this conventional charging method, a large portion of the charging occurs during the constant voltage step especially for the case of high current charging (fast charging), which further increases the charging time with a low current density. The decrease of the current density in the constant voltage region cannot be viewed as fast charging. Therefore, injection of different C-rate in different time periods during the voltage limit region somewhat prevents the researchers from comparing the effects of cell degradation caused by the high C-rate charging method. Therefore, in this study, the experiments with the SOC range setup were conducted using the high C-rate charging for the same time and current density condition as much as possible to specifically compare the effects of the high CC and PC charging methods.
Comment 3
Figure 4:
- It is hard to understand the figure. For both 4a and 4b, the voltage starts from 3.5V to 3.8V for charging, and then discharged from 3.5V to 3V for all cycles? Should be discharged from 4.2V to 3V and change from 3V to 4.2V?
Response 3
Again, I truly appreciated the comment from Reviewer 2. I have answered in the following.
We controlled the amount of current to make the SOC range from 20 to 50%. In other words, we did not control the voltage of the batteries. The voltage might stop at any moment as long as the amount of energy charged is enough to reach at SOC 50%.
The result in Figures 4 is obtained by charging the same amount of energy by 3C constant current and by 6C pulse current. In the experiment, partial charging was performed to consider only the effect of the charging method.
Charging starts at higher than 3V due to high current density and cell deterioration. Since 3Wh, which is 30% of QI, is injected, charging voltage ends before reaching to 4.2 V. We set the charging limit to the amount of energy to observe the change of cell degradation and capability when the same amount of energy is input. In addition, partial charging can prevent the charging conditions change in the constant voltage region. Therefore, the change in the charging start point voltage and the charging voltage curve with cycle only reflect the increase in internal resistance due to battery degradation. In the case of discharging, since it was not fully charged in the previous charging step, discharging starts at a point lower than 4.2V. Moreover, the decreasing discharge starting voltage is reflecting the internal resistance of the cell, not changing the experimental conditions.
Comment 4
- For 4a, it seems like the battery are not charged to the same condition, i.e. the stop charge voltage are significant differences for each cycle.
Response 4
Figure 4a was run under perfectly identical experimental conditions for all cycles. For charging, 3Wh of energy was injected at a current density corresponding to 3C. For discharging, 3Wh of energy was extracted to a current density corresponding to 1C. Since we limited the charging conditions by the current density and the amount of energy (Wh), we thought that most of the changes in the voltage curve and capacity shown in Figure 4a are due to battery degradation. This choice of charging method is because our experiment is aimed at comparing the effects of charging methods. In studies to evaluate new electrodes, electrolytes, protective films generally use full voltage windows (3.0V to 4.2V). However, such constant current-constant voltage charging is difficult to ensure justification for charging time. This is because if the cell reached 4.2V during charging due to high current density or battery deterioration, the current density decreases while charging at constant voltage, and the charging time increases. In conclusion, it is correct that we choose partial charging to maintain the current density condition, but the charge condition proceeded the same in all cycles.
Comment 5
- For 4b, based on the author’s description, the battery is charged for 0.5s and relax for 0.5s. If this is true, why each charge/relax curve is so smooth and all start from almost the same voltage?
Response 5
In Figure 4b, there are two upward-right curve groups and one downward-right curve group. Among them, the downward-right group means the voltage profile representing the constant current discharge. Among the two upward-right voltage profile groups, the upper group displayed over the range of 3.7V to 4.2V is the voltage curve in the state where current is applied (pulse charge). The lower group displayed over the range of 3.25V to 3.8V is the voltage curve at rest time. The two upward-right graphs are connected. However, since the pulse period is very short (0.5 seconds), the entire voltage curve is drawn in the form of a plane. In this case, it was difficult to grasp the change in the voltage curve for each cycle, so we deleted the connection line between the pulse region and the idle region. In Figure 4b, our opinion is that the cell voltage decreases instantly through a short pause, which relieves the load on the cell. As a result, it can be seen that even after the 150 cycles, the increase of internal resistance and decrease of capacity is suppressed than the constant current in figure 4a.
Comment 6
- In addition, the battery resistance increased after cycling, why the ohmic drop at the 150th cycle discharge curve is less than the ohmic drop at the 10th cycle discharge?
Response 6
We have identified the corresponding data points in the original text file. On the 150th, it was confirmed that one data point corresponding to the rest time after charging of the previous cycle was entered. Therefore, we have corrected the data.
Comment 7
Line 147: Why it limited to 3Wh? This paper compared the 3C-CC and 6C-PC 0.5s pulse with 0.5s relaxation. Is 3C-PC or 10C-PC will be better than 6C-PC? Will 0.2s pulse better?
Response 7
In this study, the charging energy limit of 3Wh, the constant current charging of 3C, and the pulse charging of 6C were selected as controls to make a clear comparison of the degree of battery degradation according to the charging method. We conducted various experiments with 1C, 2C, and 4C constant currents. However, since 1C and 2C are not fast, and the degradation was also slow. On the contrary, from 4C, it was difficult to observe a clear trend of performance change due to too rapid deterioration.
3 Wh was chosen to keep the charging time, implantation energy, and constant current density from a practical point of view. This is because constant voltage charging occupies a long time when the injection energy is increased or when using the fully charged method. Since this causes a decrease in the current density, it was considered unsuitable for comparing the constant current and the pulsed current.
The current density and pause time of the pulse charging was set to 6C and 0.5 seconds for simple and accurate comparison with the control 3C constant current. This is because we tried to observe the cell performance change in the situation where the same time and energy are injected. When charging with a current density of 6C in a period of 0.5 seconds, it is theoretically possible to charge the same capacity for the same time. Therefore, it is believed that the experimental group showed the possibility of suppressing battery deterioration by the charging method.

Reviewer 3 Report
Dear Authors,
Optimal charging of Lithium-Ion (Li-Ion) batteries is a challenge. Rechargeable Li-Ion batteries have a limited life. The typical estimated life of a Li-Ion battery is about three - four years or 500 - 1000 charge cycles. Contrary to appearances, battery charging is a complex process. Li-Ion batteries require appropriate charging parameters:
- Temperature - Batteries don’t tolerate overheating. Manufacturers use advanced algorithms to limit battery overheating. Exceeding these parameters results in a reduction of their service life.
- Current intensity - The 1C rule is most often used (the 1000mAh battery is charged with a current of 1A). The battery life decreases with increasing current intensity
- Frequency of charging - Today's batteries "like" frequent and short charging (<1C, few minutes).
- Overchargeable - The Li-ion battery cannot be charged above the nominal voltage. After exceeding this value, the cell is damaged.
In my opinion:
- Best to recharge the batteries to the max current 0.8 C (a lengthy process).
- Charging with max current up to 0.9 C is already harmful to the battery.
- Each short charge (>>1C) will shorten its service life.
The Authors propose to increase the charging speed of the Li-Ion battery at the expense of its lifetime. The Authors say their method minimally shortens the battery life (500/450=10%). The advantage to be significantly shorter charging time (until now, a significant disadvantage). It seems that it’s possible. It would be a small breakthrough in this industry, but today I cannot verify it in practice. However, there are seven problems:
1st Problem – The Temperature
Charging with high current translates into an increase in temperature inside the battery. This is a very dangerous process. Explain, please.
2nd Problem – The Safety
The use of a high charging current for a long time is close to the safety limit. If it is exceeded, it always leads to a fire. This aspect should be analyzed in real applications. Explain, please.
3rd Problem – The Quality
Today's knowledge of Li-Ion batteries is as follows: fast charging with very high current destroys the battery.
So there is a dilemma:
Charging speed vs power quality vs lifetime
Explain, please.
4th Problem – The Ecology and Economy
There is another dilemma:
A shorter battery life means more waste and more costs. I would like to see an economic analysis. Is it worth it?
Explain, please.
5th Problem - Measurement Uncertainty
In order to enable the comparison of these results of measurements, they should be reported (in accordance with the Guide to the Expression of Uncertainty in Measurement) along with the values of their uncertainty. The Authors ought to estimate these values, which will allow determining the „accuracy" of the proposed method and its practical utility. In the discussion section, I would expect to see an analysis of the possibilities to reduce the measurement uncertainty.
6th Problem - The Validation
Each new measurement method must be verified. It must be validated.
7th Problem – The Application
My guess is that the project is dedicated to mass use in e-cars.
But
The experiment was conducted under laboratory conditions. In a real application there is a problem of changing weather conditions and an underestimated problem of vibrations. These two parameters will definitely change the charging characteristics.
Other
And the Warburg impedance... the next paper?
Conclusion:
This article is worth discussing further.
Author Response
Response to the comments from Reviewer 3
First of all, I really appreciate careful review from Reviewer 3. Furthermore, I am so grateful for Reviewer 3 as he/she shares such valuable insights of the battery characteristics. I fully agree with those ideas.
Optimal charging of Lithium-Ion (Li-Ion) batteries is a challenge. Rechargeable Li-Ion batteries have a limited life. The typical estimated life of a Li-Ion battery is about three - four years or 500 - 1000 charge cycles. Contrary to appearances, battery charging is a complex process. Li-Ion batteries require appropriate charging parameters:
- Temperature - Batteries don’t tolerate overheating. Manufacturers use advanced algorithms to limit battery overheating. Exceeding these parameters results in a reduction of their service life.
- Current intensity - The 1C rule is most often used (the1000mAh battery is charged with a current of 1A). The battery life decreases with increasing current intensity
- Frequency of charging - Today's batteries "like" frequent and short charging (<1C, few minutes).
- Overchargeable - The Li-ion battery cannot be charged above the nominal voltage. After exceeding this value, the cell is damaged.
In my opinion:
- Best to recharge the batteries to the max current 0.8 C (a lengthy process).
- Charging with max current up to 0.9 C is already harmful to the battery.
- Each short charge (>>1C) will shorten its service life.
Based on the comments from Reviewer 3, I have prepared the answers in the following and revised the manuscript accordingly if needed also.
Comment 1
1st Problem – The Temperature
Charging with high current translates into an increase in temperature inside the battery. This is a very dangerous process. Explain, please.?
Response 1
I truly appreciated the comment from Reviewer 3. Here is my answer and opinion in response to the Comment 1
During the charging process, Li ion is being pulled into the carbon layer through the SEI. This process is dominated by the dispersion process and it really takes time. If the charging process continues at high current without some sort of relaxation, then temperature increases due to the dispersion resistance (Warburg resistance) resulting into a rapid growth of SEI layer and then again increasing the resistance even further. If this process continues and the temperature reaches to the point where it can trigger thermal runaway of the cell. It will eventually cause the disastrous fire. Therefore, it is critical to monitor the cell temperature carefully not to cause the thermal runaway. To make it better, it would be nicer if the charging process itself does not cause the significant temperature increase with an appropriately designed charge current profile. Indeed, this is the main goal of the current research.
I have revised the Concluding Remarks as follows to discuss how we addressed the problem.
In the current research, the systematic approach to develop a high current fast charging algorithm for LIB and understanding the mechanism underneath the aging phenomena scenarios were carried out. Based on the comparative results, the 6C-PC charging method that contains 1 Hz algorithm (consist of 0.5 sec charging and 0.5 sec resting) show incredible electrochemical performance over 400 cycles with the targeted commercial cylindrical battery. The pulse charging method can support for charging 30% of the battery capacity in 6 min 20 sec, with the time offset of 0.3 sec/cycle, which are much better than the traditional constant current charging method. In addition, the constant charge limit voltage (at 4.2 V) of the 6C-PC charging method implies that the method can greatly reduce the development of internal impedance. The EIS observation by cycling proves that the pulse current charging process limits the growth of the RS and the RCT, which can maintain well the lithium ion insertion and diffusion into the electrodes. Astonishingly enough, this approach provided the notable success in the fast charging with the pulse current charging. Therefore, the high-current pulse charging process of about 1 sec is, in our opinion, very effective for commercial batteries and practical applications on EV.
Comment 2
2nd Problem – The Safety
The use of a high charging current for a long time is close to the safety limit. If it is exceeded, it always leads to a fire. This aspect should be analyzed in real applications. Explain, please.
Response 2
I really appreciated the comment from Reviewer 3.
As commented by Reviewer 3, exposure to the high current charging for a long time surely increases the temperature due to the resistance and then again accelerates the SEI formation resulting into yet higher resistance build up during the charging and it gets worse as the cycle repeats. Again, it is the focus of the current research. In other words, the main goal of the research is to find a possible optimal fast charging strategy that uses high current while avoiding excessive temperature increase and excess SEI formation. The finding reported in the current study clearly shows the possible safe and fast charge strategy and reported the rationale behind the effectiveness of pulse charging strategy. Please, refer to the Response 1 as well.
Comment 3
3rd Problem – The Quality
Today's knowledge of Li-Ion batteries is as follows: fast charging with very high current destroys the battery.
So there is a dilemma: Charging speed vs power quality vs lifetime
Explain, please.
Response 3
Again, I truly appreciated the comment from Reviewer 3.
The said problem is all related to the high current charging without an appropriate relaxation in between (constant current). The CC charging profile causes the battery to be exposed for a long time causing excessive temperature rise and rapid SEI formation and eventually shortens the life time of the battery. As explained the previous responses, the main goal of the current research is to come up with the strategy to prevent the long time, continuous exposure to the high charging current. Similar efforts were made by other researchers as well and I have added a few paragraphs as shown below towards the end of 3. Results and Discussion
Similar observations were reported by other researchers as well while the viewpoints were somewhat different among each other. In the study by Li, J. et al. [18], SEM images of cathodes and anodes from the all tested cells were taken and the comparison showed that the cathodes were all in the similar conditions regardless of charging profiles whether it is a conventional CC-CV or a pulse charging profile, while the anodes showed significantly less SEI formation for the case of the pulse charging profiles. These results were nicely in line with the current study of 1 Hz pulse charging strategy which targeted specifically for minimal SEI formation during the charging. Also, in the study by Abdel-Monem, M. et al. [19], comparison study between a multi-stage constant current and constant voltage (MCC-CV), a conventional CC-CV, and a pulse charging profiles were reported. The conclusion showed that the battery deteoration were severe for the case of CC-CV while rate of capacity decrease were minor for the cases of MCC-CV and the pulse charging and somewhat similar between them. Hoewever, they also reported that the lower rate of impedance rise were clear characteristic of the pulse charging case compared to the case of MCC-CV which implied that better anti-aging performance of the pulse charging strategy. Similar results were also reported in other studies as well [20-22].
Comment 4
4th Problem – The Ecology and Economy
There is another dilemma:
A shorter battery life means more waste and more costs. I would like to see an economic analysis. Is it worth it?
Explain, please.
Response 4
I fully agree with this comment from Reviewer 3. As I answered in the previous responses, the goal is not to shorten the battery life by charging the battery with the proper fast charging strategy. Please refer to the Response 3 for findings from other researchers as well.
Comment 5
5th Problem - Measurement Uncertainty
In order to enable the comparison of these results of measurements, they should be reported (in accordance with the Guide to the Expression of Uncertainty in Measurement) along with the values of their uncertainty. The Authors ought to estimate these values, which will allow determining the “accuracy” of the proposed method and its practical utility. In the discussion section, I would expect to see an analysis of the possibilities to reduce the measurement uncertainty.
Response 5
I fully agree with this comment from Reviewer 3. When the results from the current investigation were obtained, the advantage of the pulse charging at the specified frequency were so obvious and therefore, careful attention to the measurement uncertainty was not fully exercised as pointed out by the Reviewer 3. Currently, a series of experiments are being carried to strengthen the arguments for the pulse charging strategy with certain frequencies and after the completion of the on-going study, careful report on the uncertainly of the measurement will also be included as well as the favorable effects of the pulse charging strategy. Again, I truly appreciate the comments from Reviewer 3 on this matter.
Comment 6
6th Problem - The Validation
Each new measurement method must be verified. It must be validated.
Response 6
Again, I fully agree with this comment from Reviewer 3. As I mentioned in the previous response, the advantage of the pulse charging was so obvious from the experimental results and therefore, the paper was prepared and submitted with other results from the other investigation as references. Nonetheless, I fully agree with the comments from Reviewer 3 and I DO respect that. Currently, a series of study on the pulse charging strategy with certain frequencies are on-going. With the updated results from these experiments, further validation can be provided as pointed out by Reviewer 3 in the next report.
Comment 7
7th Problem – The Application
My guess is that the project is dedicated to mass use in e-cars. But, the experiment was conducted under laboratory conditions. In a real application there is a problem of changing weather conditions and an under-estimated problem of vibrations. These two parameters will definitely change the charging characteristics.
Response 7
I value this comment from Reviewer 3 again. For real life charging condition may vary from time to time and surely changes depending on the locations. For temperature variations, it is for sure that further study has to be conducted. At the moment, users of e-cars are expected to charge the vehicle during the normal and favorable weather conditions. I hope this assumption is not far from a reasonable guess. For the vibration issue, it is less likely to be a major concern since the charging can only be done in parking unless a dynamic wireless charging is being considered. Nevertheless, it is clear that the weather condition has to be investigated further to strengthen the pulse charging strategy for sure. I truly appreciate your comments on this sensitive matter.
Comment 8
Other
And the Warburg impedance... the next paper?
Response 8
First of all, I am very grateful to have a chance to exchange ideas with Reviewer 3. As I mentioned in Response 5, 6, a series of next experimental study is being carried out. It would provide us a wealth of information that would be firmly support the pulse charging strategy with a certain frequency. At the moment, it seems quite important to related the Warburg impedance to estimate the favorable pulse charging frequency as explained in Response 1. In any case, with the upcoming results from the cases of various charging frequencies, it is fully expected to enlighten the understanding of pulse charging strategy.
I DO appreciate the comments and suggestions from Reviewer 3 and I would like to express my sincere gratitude.

Round 2
Reviewer 2 Report
The paper is improved after the modification, but still have issues as below:
- In figure 4b, the cycling issue is still there for discharging. The ohmic drop from 30th to 150th cycle discharge curve is decreasing which conflicts with the author's conclusion about SEI layer increase will increase the resistance.
- For figure 4a, the author did not explain it well and the figure still conflicts with the normal sense. Although the author did not control the voltage, the voltage for cycling of charing and discharging should be continued i.e. voltage increase from 3.5 to 3.8V during charging then drop from 3.8V to3V, then it should above 3V slightly instead of going back to 3.5V directly. This result seems like the battery is not got charge and discharged at all. It would be better to show a continued voltage curve for a few cycles.
Author Response
Response to the comments from Reviewer 2
I appreciate careful review from Reviewer 2 asking for further clarification. For two outstanding comments, I have prepared the responses for each as follows. Also, based on the comments from Reviewer 2, I have revised the manuscript accordingly as well.
Comment 1
In figure 4b, the cycling issue is still there for discharging. The ohmic drop from 30th to 150th cycle discharge curve is decreasing which conflicts with the author's conclusion about SEI layer increase will increase the resistance.?
Response 1
I do respect your careful observation and I appreciate that you gave us a chance to clarify what had happened.
For the case of old battery (let’s say the battery after 150 cycles), due to the increase of internal resistance, the charging voltage increased relatively higher (near 4.2 V). After the 5 minutes of rest time, the discharging was initiated. With this limited rest time, the voltage did not reach down to an open circuit voltage. In other words, the discharging was initiated from the high voltage level at around 3.7 V before it really reached down to the fully converged open circuit voltage. By comparison, for the case of fresh battery (let’s say the battery after 10 cycles), due to the low internal resistance, the charging voltage increased only up to 3.9 V. With the limited rest time of 5 minutes, the voltage dropped to near 3.62 V and then the discharging was initiated.
This is where the confusion comes from and if we had done the experiments with the long enough relaxation time, then we could have obtained the graphs that would behave nicely along with the opinion from Reviewer 2. Since, the main goal of the current research was to confirm that applicability of the pulse charging, we did not care to explain the details for the discharging behavior as much.
However, now, in order to respect the comment from Reviewer 2, I have revised the manuscript to include the following explanation in red color (in the middle of Page 5 before Figure 5).
One more thing to clarify in Figure 4 is the discharging voltage levels of the battery at different cycles. For the case of old battery, namely the battery after 150 cycles, due to the increase of internal resistance, the charging voltage increased relatively higher (near 4.2 V). After the 5 minutes of rest time, the discharging was initiated. With this limited rest time, the voltage did not reach down to an open circuit voltage. In other words, the discharging was initiated at the high voltage level at around 3.7 V before it really reached down to the fully converged open circuit voltage. By comparison, for the case of fresh battery, i.e. the battery after 10 cycles, due to the low internal resistance, the charging voltage increased only up to 3.9 V. With the limited rest time of 5 minutes, the voltage dropped to near 3.62 V and then the discharging was initiated. Normally, the battery with higher internal resistance shows lower discharging voltage level. But, in the current study, the discharging processes were started only after the limited rest period and therefore the voltage of the old battery came out to be higher than the relatively fresh battery since the discharging started from the higher voltage level.
Also, I have replaced the graph with color coded curve for each cycle.
Comment 2
For figure 4a, the author did not explain it well and the figure still conflicts with the normal sense. Although the author did not control the voltage, the voltage for cycling of charging and discharging should be continued i.e. voltage increase from 3.5 to 3.8V during charging then drop from 3.8V to3V, then it should above 3V slightly instead of going back to 3.5V directly. This result seems like the battery is not got charge and discharged at all. It would be better to show a continued voltage curve for a few cycles.
Response 2
In response to the comment from Reviewer 2, we showed a voltage vs time graph as follows. As shown in the graph below, there was a rest time of 5 minutes prior to the charging process and before a discharging process and so on. The graph shows the details of voltage and current profiles from the 28th discharging to the 31st discharging. The blue line is the voltage curve, and the red line is the current curve.
As clearly displayed in the graph, during the rest period after the charging, the voltage decrease was observed and during the rest period after the discharging, the voltage increase was noticed as commonly accepted findings. In the current study, the amount of charging voltage change was rather large due to a high current charging density.
In order to accommodate the comment from Reviewer 2, I have added a statement in the manuscript as following in red color. (at the end of Page 2)
A rest time of 5 minutes was given between each process, i.e. 5 minutes of rest time after the charging and 5 minutes of rest time after the discharging.
